# Passive acoustic methods for tracking the 3D movements of small cetaceans around marine structures

**Douglas Gillespie**[1]*, **Laura Palmer**[1], **Jamie Macaulay**[1], **Carol Sparling**[2], **Gordon Hastie**[1]

**1** Sea Mammal Research Unit, Scottish Oceans Institute, University of St Andrews, St Andrews, Scotland,
**2** SMRU Consulting, Scottish Oceans Institute, University of St Andrews, St Andrews, Scotland

* dg50@st-andrews.ac.uk

**Data Availability Statement:** Click detection data and the vessels location from the reported calibration trial are available at https://doi.org/10.

## Abstract

A wide range of anthropogenic structures exist in the marine environment with the extent of these set to increase as the global offshore renewable energy industry grows. Many of these pose acute risks to marine wildlife; for example, tidal energy generators have the potential to injure or kill seals and small cetaceans through collisions with moving turbine parts. Information on fine scale behaviour of animals close to operational turbines is required to understand the likely impact of these new technologies. There are inherent challenges associated with measuring the underwater movements of marine animals which have, so far, limited data collection. Here, we describe the development and application of a system for monitoring the three-dimensional movements of cetaceans in the immediate vicinity of a subsea structure. The system comprises twelve hydrophones and software for the detection and localisation of vocal marine mammals. We present data demonstrating the systems practical performance during a deployment on an operational tidal turbine between October 2017 and October 2019. Three-dimensional locations of cetaceans were derived from the passive acoustic data using time of arrival differences on each hydrophone. Localisation accuracy was assessed with an artificial sound source at known locations and a refined method of error estimation is presented. Calibration trials show that the system can accurately localise sounds to 2m accuracy within 20m of the turbine but that localisations become highly inaccurate at distances greater than 35m. The system is currently being used to provide data on rates of encounters between cetaceans and the turbine and to provide high resolution tracking data for animals close to the turbine. These data can be used to inform stakeholders and regulators on the likely impact of tidal turbines on cetaceans.

## Introduction

Anthropogenic structures have increased in number in the marine environment over the past several decades with increases in oil and gas exploration and extraction, marine aquaculture, and renewable energy [1]. Many of these activities may pose acute risks to marine wildlife; for example, seabirds can be killed by wind turbines [2] and cetaceans can be injured or killed as a

17630/de341ca6-6754-43ed-b1ac-a55aae6ccfaa.
Other data have been included in the manuscript.

**Funding:** This research was funded through a research grant from the Scottish Government as part of the Marine Mammal Scientific Support Program MMSS/002/15. Representatives of the Scottish government and Marine Scotland sat on the project steering committee, overseeing experiment design, and also provided feedback on early drafts of this paper.

**Competing interests:** The authors have declared that no competing interests exist.

result of vessel collisions [3] and fisheries bycatch [4]. Many countries are now also looking to generate low carbon electricity through the installation of underwater turbines in areas of high tidal flow. A common design, horizontal-axis turbines, broadly resemble small wind turbines mounted on the sea floor. Just as the prevalence of wind farms has raised concerns about the risk to birdlife, tidal turbines have the potential to injure or kill animals through collisions with moving rotors [5]. Several taxa are considered to be vulnerable to these risks, including diving birds, fish, and marine mammals [6].

To understand interactions between animals and structures, information on under-water movements of animals around them is required. A number of technologies including video, infra-red based detection, and radar have been used to detect and track birds and bats around windfarms to inform estimates of collision risk [7]. However, there are inherent challenges associated with measuring the underwater movements of marine animals, particularly in highly energetic and turbid environments. Movement tags have been used to study the under-water behaviour of marine mammals, particularly in response to underwater sound [8,9]. However, difficulties in deploying and recovering tags, limited deployment periods (days or weeks), and the fact that a tagged animal may never visit a structure of interest, mean that for many species tagging programs are unlikely to yield fine scale data at very specific locations.

Small cetaceans are highly vocal, using echolocation clicks to actively sense their environment [10]. Harbour porpoises (*Phocoena phocoena)* produce narrow-band high-frequency clicks with a centre frequency of ~130kHz and a duration of ~77μs [11,12]. Most delphinid species produce broader band echolocation clicks with energies mostly between ~30 and over 100kHz [10]. Unlike harbour porpoise, dolphins also produce whistles which can be highly variable but are mostly between around 5 and 20kHz with durations of less than a second [13,14]. Arrays of hydrophones can be used to detect and locate cetaceans underwater [15,16] and it is possible to track the movements of cetaceans in the vicinity of anthropogenic structures [17].

Here we describe a hydrophone system to detect, classify and localise individual cetaceans with a high degree of spatial and temporal accuracy. We describe both the hardware and software and discuss the key principles and limitations when using hydrophone arrays to localise cetaceans in three dimensions (3D). We then report on the practical performance of the system through a study where it was deployed on an operational commercial-scale tidal turbine.

## Materials and methods

To collect data on the 3D movements of harbour porpoises and other small odontocetes around anthropogenic structures, we designed and built a hydrophone array, and acoustic acquisition and processing system. We then deployed the system and collected data over a two-year period between October 2017 and October 2019. The system was semi-automatic, with real time detectors reducing amounts of stored data by several orders of magnitude, but operator screening of remaining sounds required to select and confirm detections from the stored data.

All procedures and data collection were approved by the University of St Andrews School of Biology Ethics Committee (Reference number SEC18014).

### Hydrophone array

In principle, four widely spaced hydrophones can provide 3D locations based on time of arrival differences of signals at the four receivers [15]. However small errors in timing estimation can introduce ambiguities and large errors, particularly outside the immediate vicinity of the array. Detecting highly directional sounds on widely spaced hydrophones can also be

problematic since if an animal is orientated towards one hydrophone, it will generally not be orientated towards the others. Further, echolocation clicks are often produced with short inter-click-intervals (ICI) which makes matching corresponding clicks between hydrophones challenging with widely spaced hydrophones. Small clusters of hydrophones, where the maximum sound travel time between the hydrophones is small compared to typical inter-click intervals, arranged in a tetrahedral pattern, do not suffer from matching ambiguities but can only measure horizontal and elevation angles to a sound source, not range. For 3D localisation, a practical compromise is to deploy multiple small tetrahedral clusters of four hydrophones. Each cluster can measure angles to sound sources and, when a sound is detected on two or more clusters, the sound can be localised in 3D.

Twelve hydrophones (in three clusters), each consisting of a 10mm ceramic sphere with 0.8mm wall thickness (S10 hollow sphere from Yujie Technology Ltd, Qunxing-Square, Huaqiang North Road, Futian district, Shenzhen, Guangdong, China 518028) were potted in ALH Systems NP1480 water clear polyurethane, shaped to mount in the ends of 10 mm diameter steel tubes epoxied into an acetyl base to provide a 15cm spacing between each hydrophone (Fig 1 inset). Hydrophone sensitivity was calibrated against a Reson TC4013 reference hydrophone and was measured to be -210±1 dBre1V/µPa. Short (<20cm) cables from each hydrophone carried signals inside the mounting tubes to a box containing custom built preamplifiers with a gain of 30dB. For robustness, the preamplifiers and twisted pair signal and power cable were solid potted into this box using epoxy. The hydrophone supports were bolted to 1cm x 80cm x 80cm high density polyethylene sheets, which could in turn be bolted to other structures.

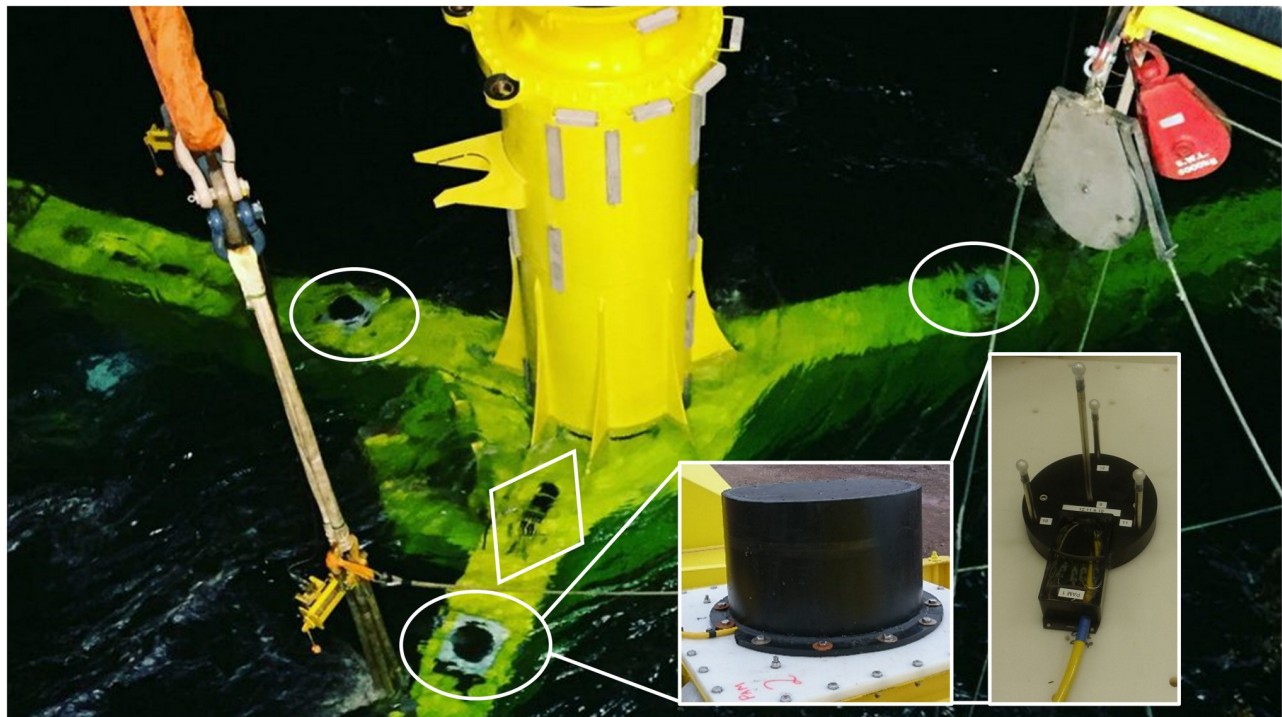

**Fig 1. Photograph of three hydrophone clusters installed on a turbine support structure during installation showing the locations of the three hydrophone clusters (circled) and acquisition junction box (in diamond).** Inset are photographs of a hydrophone cowling and a tetrahedral cluster showing the four hydrophones on their supports and the box containing potted preamplifiers (photo SIMEC Atlantis Energy).

To protect the hydrophones from physical damage, each cluster was covered with a 50cm diameter and 50cm high 10mm thick polyethylene cowling. High density polyethylene was chosen as it has an acoustic impedance of 2.33 x10$^5$gm/cm$^2$.sec (https://www.ndt.net/links/proper.htm), similar to sea water (impedance 1.5x10$^5$gm/cm$^2$.sec). This means that the reflection coefficient of a sound in water hitting the polyethylene will be approximately 0.21, or -13.3dB.

## Data acquisition and real time processing

Analog signals from hydrophone clusters were brought to an underwater junction box containing the signal conditioning and data acquisition system. Preamplifiers (ETEC 150410AQC, Etec aps. Industrivaenget 8, DK-3300 Frederiksvaerk, Denmark) provided a further 20dB of gain and had two-pole high and low pass filters at 3.1kHz and 190kHz respectively. Outputs of the preamplifiers connected to three NI-9222 four channel simultaneous sampling 16-bit acquisition modules contained in a CRIO-9067 controller (National Instruments Corporation, Austin, TX 78759, USA). The CRIO was programmed to acquire data continuously on all channels at a sample rate of 500kS/s per channel. Data were compressed in real time using lossless compression [18] and sent via Ethernet to a shore-side PC for processing.

On shore, real time processing was conducted using the open source PAMGuard software [19]. A bespoke acquisition module was written to control the CRIO system, unpack the compressed audio data and insert it into the PAMGuard processing chain. A separate "watchdog" program was written which started automatically if the system rebooted (e.g. after a power outage) and would start or restart PAMGuard in the event of data processing being interrupted for any reason.

## Offline analysis

**Click classification and event selection.**   The click detector in PAMGuard was configured to trigger on any transient sound with energy rising more than 10dB above background noise in the 40 to 150kHz frequency band. It made large numbers of false positive detections as well as cetacean echolocation clicks (true positives). False positives can be caused by a variety of sources including flow noise over the structure and hydrophone mounts, operational noise and passing vessels. Transient signals were classified as porpoise clicks if they were between 20μs and 220μs duration, had a peak frequency between 100 and 150kHz and had total energy in the 100 to 150kHz frequency band at least 6dB higher than both the 40 to 90kHz and the 160 to 190kHz bands. Although harbour porpoise clicks are narrow-band and distinct from the other noises, broadband dolphin clicks can be challenging to distinguish from background transients. Key to identifying echolocation clicks is often their occurrence at regular intervals on a consistent bearing. Data were therefore screened manually post hoc using PAMGuard, primarily viewing displays of bearings to sounds derived from the individual hydrophone clusters plotted against time. This follows the methods described for sperm whales in [20], whereby the analyst assigns clicks to 'events' based on their consistency of bearing and other properties (waveform, power spectra, amplitude and inter click intervals).

**Localisation and associated errors.**   Three-dimensional click localisation was a multistage process which, although largely described in [16], had a series of modifications to the ways in which errors on localisations were calculated. Sounds detected on more than one hydrophone cluster can only originate from the same source if the time of arrival difference between those sounds is less than the distance between the clusters divided by the speed of sound; in this case a maximum of 7.4ms. Although small cetaceans generally produce clicks with inter click intervals (ICI's) greater than this, ICI's in porpoises can be as low as 2ms [9]

which can cause ambiguities in matching clicks between hydrophone clusters. To resolve this 'click match ambiguity', clicks from the different hydrophone clusters were initially divided into groups consisting of clicks arriving within 7.4ms of a click a different cluster. Hence, clicks within a group might originate from the same sound but it was impossible that any of them matched with clicks in other groups [16]. All possible combinations of clicks within each group were then localised and scored according to the number of clusters used and the quality of the localisation. Combinations using all three hydrophone clusters were given a higher score than combinations on only two hydrophone clusters. Combinations using the same number of clusters were scored according to a log likelihood value from the localisation (see below). The highest scored localisation from the group was selected for further analyses.

Clicks detected on multiple hydrophone clusters were localised by measuring the Time of Arrival Difference (TOAD) of the signal on each hydrophone pair [21] and then maximising a likelihood based model comparing expected TOADs for putative locations with measured TOADs. For each possible group of clicks, a TOAD was measured for every possible hydrophone pair. This means that for a click detected on all three clusters, there would be 66 different time measurements.

Localisation estimates are subject to uncertainties caused by errors in the estimation of the TOADs as well as errors in the hydrophone locations and the speed of sound in water. The contributions of these different errors also vary depending upon which hydrophones are used in each measurement and the position of the localisation relative to each hydrophone pair; for example, a TOAD measured at a point in line with a pair of hydrophones will be directly affected by an error on the sound speed measurement, whereas that error will contribute nothing if the sound source is equidistant from the two hydrophones. Estimates of the various error components are shown in Table 1.

The total expected timing error $\sigma t_j$ for each hydrophone pair is

$$\sigma t_j = \sqrt{(var_t + var_g + var_c)} \tag{1}$$

where $var_t$, $var_g$ and $var_c$ are the variances due to TOAD timing error, geometry error and speed of sound error.

The variance $var_t$ due to TOAD estimation is calculated as the square of the estimated timing error from Table 1.

The variance contribution for geometry errors, $var_g$ is dependent on the relative positions of the hydrophones and sound source. If **p** is the position vector of the sound source and **h**$_j$ are the locations of the hydrophones, having estimated errors on each Cartesian coordinate $\boldsymbol{\delta h_{ji}}$, (using the error within clusters if the hydrophones are in the same cluster or the between clusters error if the hydrophones are in different clusters) and **e**$_i$ are unit vectors along the x, y and

Table 1. Estimates of the different contributors to localisation errors.

| Source of Error | Magnitude |
| --- | --- |
| Timing Error Estimate within a hydrophone cluster | 1 μs |
| Timing Error Estimate between hydrophone clusters | 10 μs (differences in the waveform are likely to be greater between clusters making it likely that there will be a larger error) |
| Location of hydrophones within a cluster | 1mm in each dimension |
| Relative location of clusters | 5cm in each horizontal dimension, 2cm vertically |
| Speed of sound | 10m/s |

z axis, then the variance in the time difference of arrival for a hydrophone pair will be

$$var_g = \sum_{j=1}^{2} \sum_{i=1}^{3} \left( \frac{\delta h_{ji} (\widehat{\mathbf{h}_j - \mathbf{p}}) \cdot \mathbf{e_i}}{c} \right)^2 \qquad (2)$$

where $c$ is the speed of sound.

Finally, if $\delta c$ is the error in the estimated speed of sound then

$$var_c = \left( T \frac{\delta c}{c} \right)^2 \qquad (3)$$

Where $T$ is the expected time delay, i.e.

$$T = \frac{\|\mathbf{p} - \mathbf{h}_1\| - \|\mathbf{p} - \mathbf{h}_2\|}{c} \qquad (4)$$

Once the errors for each TOAD measurement for a candidate localisation have been estimated, it is possible to calculate a log likelihood estimation for that position, where

$$L(x; T) = \sum_{j} \left( -\frac{1}{2} \log \left( 2\pi \sigma t_j \right) - \frac{(TM_j - T)^2}{2\sigma t_j^2} \right) \qquad (5)$$

Where $TM_i$ is the measured time delay between the j$^{th}$ hydrophone pair. Note that many Log Likelihood estimators drop the first term in Eq 5 since it is constant. Here however, $\sigma t_j$ varies with location, so the term is retained.

The log likelihood function for each localisation was maximised using a Simplex algorithm [22]. To avoid the problem of the optimisation function finding local rather than the true maximum, four random starting points were selected for each localisation, with the first start point being the centre of the array and the subsequent ones being offset by a random distance in each dimension using a number drawn from a Gaussian distribution with width equal to the arrays maximum aperture.

Bearings to the sound source remain accurate with distance so long as there is a greater than ~10dB signal to noise ratio (SNR) to determine accurate TOAD's [21]. However, estimates of the range to sound sources become increasingly inaccurate with distance from the array. Localisation errors were therefore calculated as a range error, aligned with a unit vector from the centre of the hydrophones to the localisation, a horizontal direction perpendicular to the first and a third direction perpendicular to the other two (Fig 2), these latter two effectively being the horizontal and vertical bearing accuracies.

Errors on localisations were estimated from the curvature of the log likelihood function around its maximum value. The error estimator assumes that the overall log likelihood function follows a normal like distribution of the form

$$L_{tot}(\mathbf{p}) = -\frac{1}{2} \left( \log \left( 2\pi \sigma_{\mathbf{p}} \right) + \frac{\|\mathbf{p} - \mathbf{p}_0\|^2}{\sigma_{\mathbf{p}}^2} \right) \qquad (6)$$

Where $\mathbf{p_0}$ is the best estimated position and $\sigma_{\mathbf{p}}$ is the error on $\mathbf{p}$. It is therefore straightforward to estimate $\sigma_{\mathbf{p}}$ in any direction by slightly varying the position $\mathbf{p}$ in each direction to calculate changes in $L_{tot}(\mathbf{p})$ and then inverting Eq 6 to obtain an estimate of $\sigma_{\mathbf{p}}$.

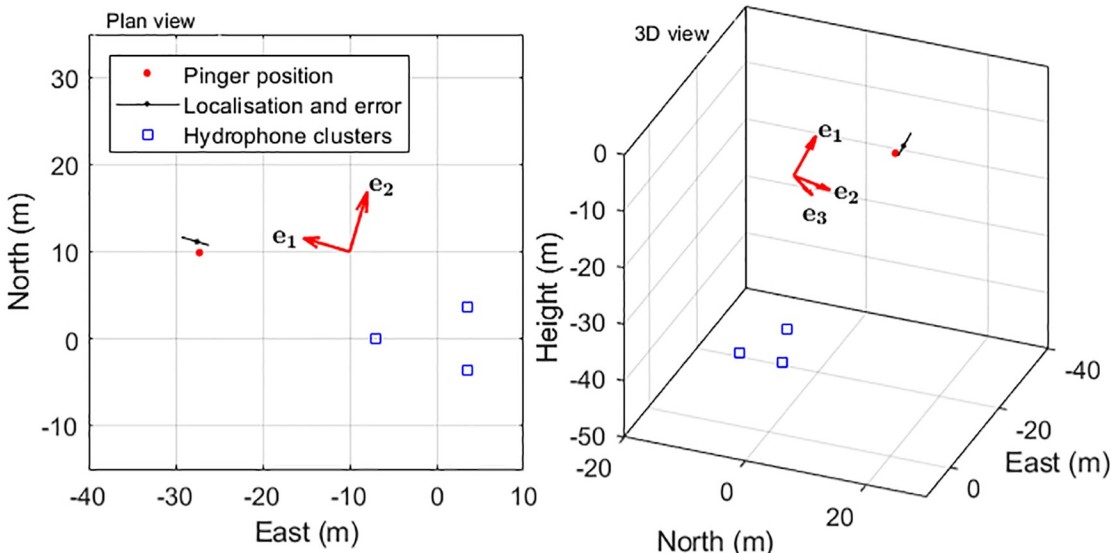

**Fig 2. Localisation error coordinate frame.** The figure shows the localisation and error estimate of a single click from the pinger trial (see below) which was detected on the north and the west hydrophone cluster, but not the south one. The largest error is in the estimation of range $e_1$, so this is aligned with a unit vector from the centre of the array. The other (smaller) errors $e_2$ and $e_3$ are aligned with the horizontal and vertical angle measurements.

**Localisation filtering.** For each localisation a $\chi^2$ value was also calculated as

$$\chi^2 = \sum \left( \frac{(TM_j - T)^2}{2\sigma t_j^2} \right) \tag{7}$$

Generally, if the clicks are correctly matched between each hydrophone cluster and the SNR of detected clicks is sufficient for accurate timing estimation, the $\chi^2$ value should be approximately equal to the number of degrees of freedom (i.e the number of TOAD measurements used minus the number of spatial dimensions, i.e. 3). However, if an incorrect combination of clicks is localised, then it is likely that the $\chi^2$ value will be markedly higher. It is therefore possible to reject erroneous localisations based on the magnitude of their $\chi^2$ value.

## Deployment on an operational tidal turbine

The 12 hydrophone system was deployed on an operational tidal turbine in the Pentland Firth (58°39'N 3°08'W), Scotland in 2016. The turbine was part of an array (the MeyGen project) comprised of three Andritz Hydro Hammerfest HS1000 turbines and the one Atlantis Resources AR1500 (https://simecatlantis.com/). The hydrophone system was integrated into the Atlantis AR1500 turbine which has a three-bladed 18m diameter rotor rotating at speeds up to 14rpm. A yaw mechanism maintained the turbine in a position facing the current, which reaches speeds up to 5ms$^{-1}$ on both the flood and the ebb tides.

The deployment was governed by the logistical constraints of installing equipment close to large and complex machinery in a highly energetic environment. Health and safety concerns made it undesirable to use divers during installation so all array hardware was attached to the Turbine Support Structure (TSS), which is a large (25 x 19m) three-legged steel frame that gravity mounts on the sea floor, prior to installation (Fig 1).

One hydrophone cluster was installed on the upper surface of each leg (Fig 1), each cluster being as far from the turbine centre as practically possible. The cluster on the westerly leg was

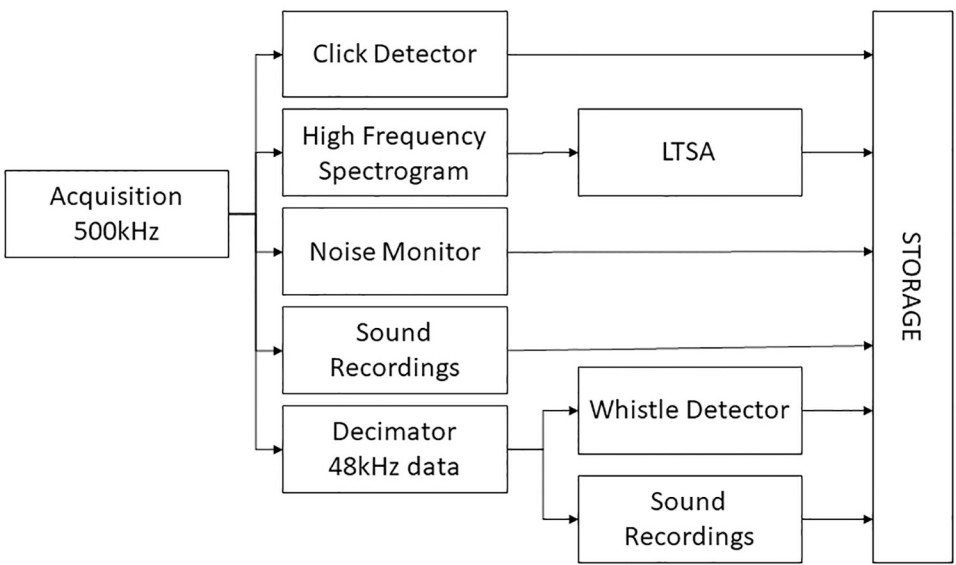

**Fig 3. Block diagram of PAMGuard processing chain.**

7m horizontally from the turbine centre, the clusters on the northeast and southeast legs were 3.5m east and 3.5m north or south. The overall array dimension was therefore 10.5m (east-west) by 7m (north-south). Hydrophones were mounted at a mean height of 12.3m below the centre of the turbine axis, i.e. 3.3m below the turbine blades and approximately 3.5m from the sea floor.

Cabling to the junction box containing the signal conditioning and acquisition system was terminated with dry mate subsea connectors. After deployment, a 48V DC power and CAT6 Ethernet cable from the junction box was plugged into the turbine's Connection Management System (CMS) using an ROV wet-mate connector. Both the CMS and the turbine nacelle can be removed for maintenance, but it is intended that the TSS (and hence the monitoring equipment) will remain in situ for 25 years. The monitoring equipment received power from the turbine's auxiliary power supply, so it was only possible to operate the system with the turbine in place.

Several different PAMGuard modules were configured for the detection of echolocation clicks, dolphin whistles and diagnostic noise measurements. A schematic of the processing chain is shown in Fig 3. To reduce the CPU load, the click detection trigger was only run on the topmost hydrophone in each cluster, but in the event of a trigger, a short clip of data from all four hydrophones in that cluster would be stored starting 100 samples (equivalent to 0.2ms or 30cm travel distance) prior to the trigger and ending 100 samples after the end of the trigger to ensure signal capture on all hydrophones in that cluster. The high frequency data (500kHz) were also decimated to 48kHz and passed to the PAMGuard whistle detector [23] to detect dolphin whistles.

The click detector trigger recorded an instantaneous measure of its background noise measurement once per second, which gives a direct assessment of the click detectors absolute threshold in varying noise conditions. In addition, the spectrogram of the 500kHz data was calculated with a Fast Fourier Transform (FFT) length of 1024 samples (2ms) and an advance of 20480 samples (40ms) and the power of the FFT's averaged and stored every 6 seconds by a Long-Term Spectral Average (LTSA) module. Octave band noise measurements in the frequency range 1.4kHz to 181kHz were also made from the top hydrophone in each of the three

clusters, averaging and storing data every 10s. Sound recording modules could store either or both of the high frequency 500kHz data or the decimated 48kHz data in wav file format. Recordings could be made either continuously or on a programmed schedule.

**Operation.**   Storing raw audio data from the 12 hydrophones in uncompressed files would have required one Terabyte of storage per day. Therefore, high frequency acoustic data were only stored during the first month of operation while detector settings were adjusted. Following this period, a 10s long, 500kHz sample rate recording was made every hour for diagnostic purposes. The primary data output from the system was from the detectors and noise monitoring modules described above. These were stored in files on the onshore computer's hard drive in a proprietary binary data format. The size of these files varies depending on numbers of detections and levels of operational noise but were typically around 3 Gigabytes per day.

Bi-weekly checks of the system were made using remote desktop software. The network connection was insufficient for data transfer, so data were copied to portable hard drives and sent by post to our lab once per month for permanent storage and further analysis.

### Localisation trial

To measure the localisation and error estimation accuracy of the system, a series of porpoise like sounds were played back to the system from a drifting vessel using the system described in [16]. Localisations from sounds detected on the hydrophones were directly compared with the known vessel position and depth of the sound source. For safety reasons, close approaches to the turbine were made when the turbine was not rotating and the blades were stationary in the 'Y' position to minimise the maximum tip height in the water column. Drifts were made with the sound source above the blade tips at a depth of either 10 or 15m. The vessel was located using a GlobalSat BU-353-S4 WAAS enabled Differential GPS with an error of less than 3m 95% of the time. Drifting with the current meant that the sound source should be directly beneath the vessel; however, due to water currents and wind action it was possible that the sound source was offset from the vessel location by several metres.

## Results

The system was deployed on 24[th] October 2016. Due to problems with the power supply to our equipment, data collection was only able to start on 19[th] October 2017 and then continued until the turbine was removed for maintenance on 22[nd] September 2018. During this initial 338 days of operation, the PAM system was operational for 322 days. Twelve days were lost due to power being unavailable either at the turbine or in the sub-station, and four days were lost due to software or computer failures. Continuous PAM data collection resumed on 18[th] December 2018 and continued until 15[th] October 2019. After the first month of operation, one hydrophone became notably noisier than the other eleven. We believe that this is electrical noise, probably caused by one half of a differential amplifier input becoming disconnected. Although some louder signals are still visible on this hydrophone, only the other eleven hydrophones were used in localisation.

A typical set of click waveforms recorded on the three hydrophone clusters is shown in Fig 4. Echoes of the signals are visible on several channels, typically occurring between 100 and 200μs after the initial signal. This delay is consistent with echoes from objects between 7.5 and 15cm away so probably came from the hydrophone mounting structures.

### Localisation accuracy

Fig 5 shows the horizontal range and vertical localisation accuracy for porpoise-like sounds measured during the calibration trial as a function of horizontal distance from the turbine

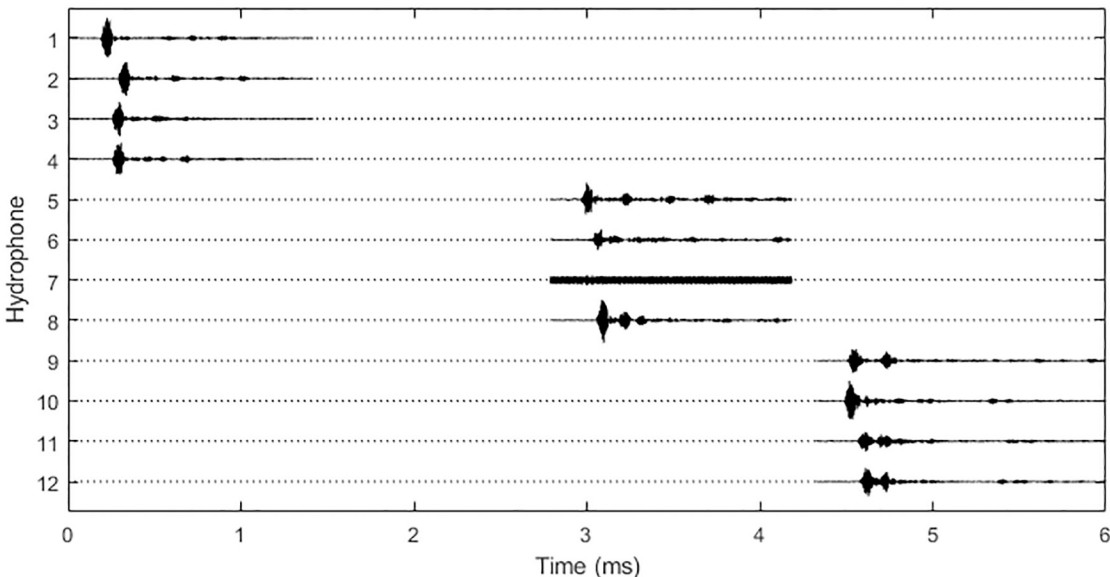

**Fig 4. A typical harbour porpoise click detected on all three hydrophone clusters.** Dotted lines show the regions where no data are recorded for each channel and solid lines show the short sections of stored waveform. Small time delays between signal arrivals within each hydrophone cluster and larger delays between signal arrivals on the different clusters are clearly visible (hydrophone 7 is not working). This group of sounds was localised to a position approximately 15m E and 9m N of the array centre.

centre. Within 20m of the turbine, horizontal accuracy was within 1 or 2m. Vertical accuracy is poorer than horizontal accuracy with errors of up to 5m even close to the turbine. Beyond a horizontal distance of 35m, both horizontal and vertical localisation errors increased significantly to 10's of metres. During the trial the sound source was above the blades, at least 24m above the hydrophones, so it is likely that localisation errors would be smaller within the area swept by the rotors, which is closer to the hydrophones.

## System noise

Noise within the system varied with both the tidal flow and with turbine operations. Fig 6 shows power spectral data over one day of operation along with individual power spectra for a

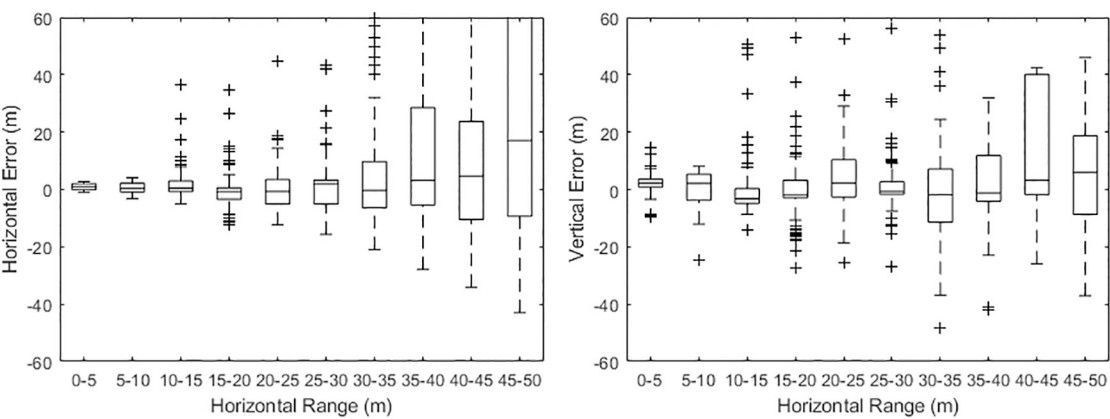

**Fig 5. Measured range and depth errors as a function of horizontal distance.**

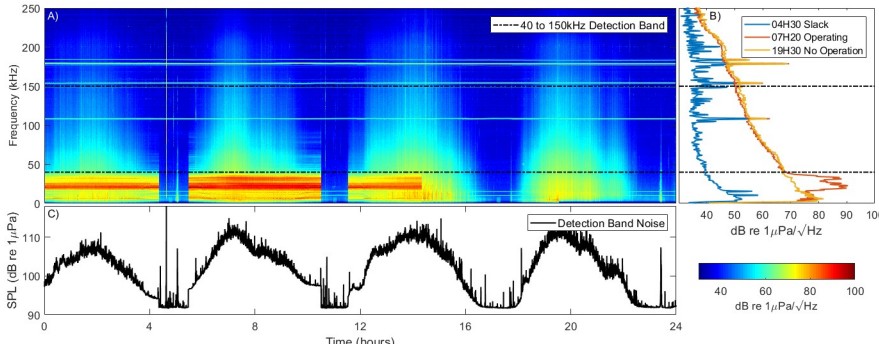

**Fig 6. System noise over a 24 hour period.** Panel A shows spectral data over the 24 hours. The four periods of tidal flow are clearly visible. Panel B shows the power spectral density at three different times during the cycle. Panel C shows the noise in the click detector detection band. The turbine operated up until 14:20 at which point generation was paused. Clearly visible are elevated noise levels at high flows over a wide frequency range. When the turbine is operating an additional noise source is evident at 20 kHz.

time when the turbine was off and a time when it was generating. Also shown is the noise in the 40–150kHz click detection band. There is an increase and decrease in noise over a wide frequency range corresponding to the increasing and decreasing tidal flow, with levels being approximately 20dB higher at peak flow compared to slack water. We believe that this noise is caused by water flow over the turbine and hydrophone structures. An additional noise is present at around 20kHz that differs from the flow noise: it has a sudden start and end and it's amplitude is constant despite changes in flow speed. It is believed that this noise is caused by electromagnetically excited vibrations ('coil whine') in the turbine generating system [24]. From phase differences in the 20kHz signal on different hydrophones, it is apparent that this noise is mechanical and not electrical.

## Animal detections

During the initial 322 days of data collection (October 2017 –September 2018), over 740 million transient sounds were recorded on the three hydrophone clusters. During offline analysis, 1044 porpoise and 31 dolphin events were marked by the analyst, containing 115380 and 57077 individual clicks respectively. Of these events, 724 porpoise and 26 dolphin events had 10 or more clicks (Table 2). Of all detected transients, only 0.02% were considered to be porpoise or dolphin sounds. Numbers of porpoise clicks per event varied considerably with a mean of 220 (95% CI 31–979). Similarly, durations of events varied from 0.5 to over 2,700 s (95% CI 21–1,200 s). It is likely that some of these events contained more than one animal.

Daily encounter rates of both harbour porpoises and dolphins varied markedly by month, with both being low in the summer months and higher in autumn and winter (Fig 7). Many events yielded few localisations, with the similarity of bearings from the different hydrophone clusters indicating that the animals were passing at distances too great for accurate localisation. Fig 8 shows the track of a porpoise passing close to the turbine. Work is ongoing to further analyse the location data in order to understand the fine scale movements of animals close to the turbine.

## Discussion

This study has shown that the PAM system described here can detect and localise small cetaceans within tens of metres of anthropogenic structures with a high degree of spatial accuracy. Calibration trials demonstrated that the system was capable of tracking high frequency sounds

**Table 2. Porpoise and dolphin encounter rates for the first year of monitoring.** Numbers in parenthesis for the numbers of animals per day are the standard error on the mean encounter rate calculated from the variance in the daily counts for each month divided by the number of monitoring days.

| Month | Days operational | Porpoise encounters | Dolphin encounters | Porpoise / day | Dolphin / day |
|---|---|---|---|---|---|
| Oct 2017 | 12.26 | 35 | 5 | 2.9 (0.9) | 0.41 (0.14) |
| Nov 2017 | 27.03 | 95 | 4 | 3.5 (0.6) | 0.15 (0.07) |
| Dec 2017 | 30.99 | 126 | 1 | 4.1 (0.5) | 0.03 (0.03) |
| Jan 2018 | 28.89 | 122 | 1 | 4.2 (0.5) | 0.03 (0.04) |
| Feb 2018 | 27.79 | 46 | 0 | 1.7 (0.2) | 0.00 |
| Mar 2018 | 25.53 | 47 | 0 | 1.8 (0.4) | 0.00 |
| Apr 2018 | 28.39 | 39 | 0 | 1.4 (0.2) | 0.00 |
| May 2018 | 30.92 | 19 | 0 | 0.6 (0.2) | 0.00 |
| Jun 2018 | 28.78 | 19 | 0 | 0.7 (0.1) | 0.00 |
| Jul 2018 | 30.36 | 46 | 1 | 1.5 (0.2) | 0.03 (0.04) |
| Aug 2018 | 30.62 | 54 | 4 | 1.8 (0.3) | 0.13 (0.08) |
| Sep 2018 | 20.26 | 76 | 10 | 3.8 (0.8) | 0.49 (0.16) |

with approximately 1-2m accuracy within 30m of the array. By connecting to power and optical fibres in the turbines export cable, it was possible to operate the system for two years with a high level of reliability and low level of operator intervention (>99% up time when power was available). The longevity of the deployment has led to the collection of 1044 porpoise and 31 dolphin events over a 12 month period.

While our results show that the system can accurately localise cetacean clicks, few whistles were detected during the deployment and we have made no attempt to localise the few that

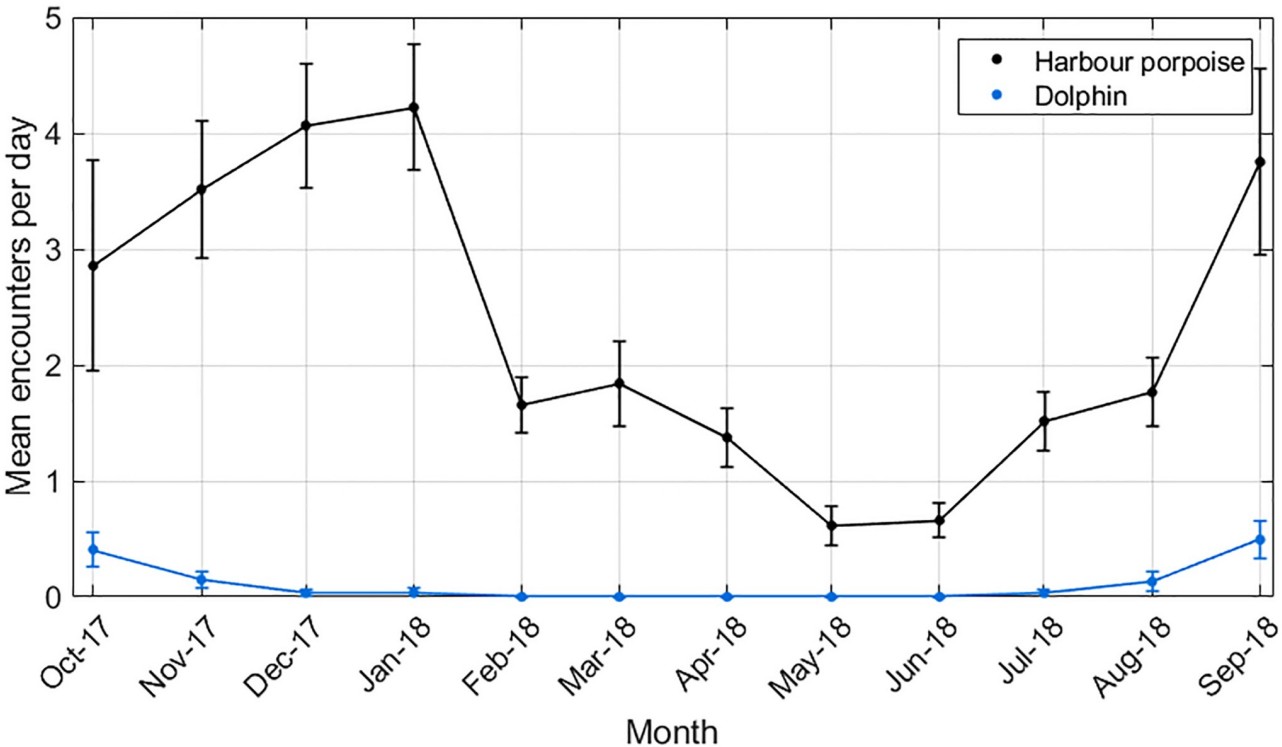

**Fig 7. Dolphin and porpoise detections by month over a one year period.**

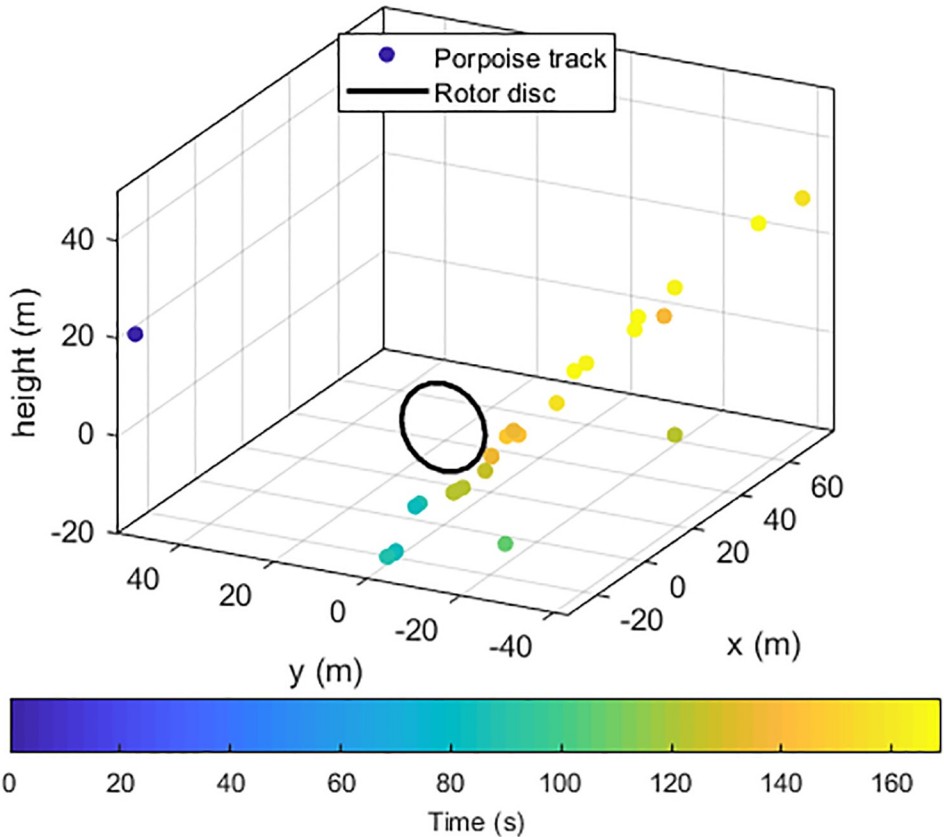

**Fig 8. Example track of a porpoise passing below the disc swept by the turbine rotors during a 160 second encounter.** Points are coloured by time. Note the three points that do not lie on the clear animal track, which are probably caused by mismatched clicks on the three hydrophone clusters. At the time of this event, the current speed was .3m/s, the turbine was rotating, but not generating.

were detected. In principle, time of arrival differences between hydrophones for whistles can be extracted [25,26] and could then be used in the same 3D localisation algorithm. However, extracting time delays from these lower frequency sounds to the same level of accuracy as can be achieved for clicks, particularly in higher noise levels at lower frequencies, may not be possible in which case the same levels of 3D localisation accuracy would not be achievable for whistles. Detecting and localising whistles during periods of tidal flow and turbine operation, when noise levels are high in the whistle frequency band would be extremely challenging.

It is also important to consider the limitations to the range at which cetaceans can be detected. This is a function of background noise, which varies considerably with environmental conditions or anthropogenic activities. Here, noise varied by 20dB over the tidal cycle and with turbine operation, meaning that cetacean detection probability would be lower during peak tidal flows. This is important here because high flow periods coincide with turbine rotation and hence when there may be risk of collision.

We do not know from these data whether the noise is transmitted to our hydrophones through the steel turbine support structure and how much of the noise is present in the water column, which might alert animals to the presence of the turbine. Measured in-water noise levels around the turbine using drifting hydrophones showed that the 20kHz noise present in our data is higher than ambient noise levels out to a range of 200m and that lower frequency sounds generated by the turbine are 5dB above ambient levels over 2km from the turbine [27].

The overall false positive rate of the click detection system was very high; only 0.02% of detected transients were retained after manual screening. This is unsurprising since the detectors were configured to reduce stored data to a manageable quantity rather than to classify with high precision. The task of extracting events took approximately one or two days manual processing for each week of data. To streamline this process for future studies, the annotated data from this first year of data collection could be used to develop automated encounter detection processes which could either be implemented to run offline in or real-time. Real-time processing has the advantage of further reducing the amount of stored data and could also provide the basis of a real-time detection system if required. For example, an ability to detect porpoises in real time around fish farms would allow for seal deterrents to be turned off when porpoises are present, allaying concerns about the effects of acoustic deterrents on porpoises [28]. The system described here was developed to determine whether or not tidal turbines pose a risk to cetaceans or whether animals will naturally avoid them. Should mitigation, such as shutting down turbines when animals approach be required, the time available to shut down a turbine following the localisation of an approaching cetacean might be less than 10s. Implementing fully automatic detection systems across all turbines in a large array, which may number in the hundreds, and engineering a braking system that could repeatedly stop a turbine in that time without regular maintenance would be both challenging and expensive.

The development of these methods is important in order to understand the potential risks associated with tidal turbines. Current collision risk models [29] are dependent on both broad-scale avoidance and fine-scale evasion of operating devices. While some studies have shown broad scale avoidance [30], there are currently no published studies describing fine scale evasive behaviour close to moving turbine blades. The techniques described in the current study should provide the means to address this data gap for harbour porpoises and potentially other small cetaceans.

Although the study described here deployed the system on a tidal turbine, similar arrays could also be used to study fine scale movement of vocalising animals around a range of other anthropogenic activities such as fish farms, ports, or other energy exploration and extraction devices during both construction and operation. We used three tetrahedral clusters of hydrophones since it suited the geometry of the available mounting structure and reduced cabling and underwater connectors. However, the localisation methods can be applied to hydrophones in almost any configuration so long as they are distributed about the volume of interest and they are spatially close enough to each other that sounds are likely to be received on a sufficient number of receivers. Optimal spacing may also vary for different cetacean species given differences in vocal behaviour and sound characteristics.

Passive acoustic monitoring alone is unable to localise animals which cease vocalising and is less likely to detect animals orientated away from the structure. Classification and tracking accuracy may be improved through the integration of other sensor systems on the platform. For example, high frequency multi-beam sonar has proven to be highly effective for the detection and classification of small marine mammals; integrating this would potentially provide a means of detecting and tracking species that vocalise infrequently or not at all [31].

In conclusion, the results presented here show that arrays of hydrophones are an effective means of detecting and tracking small cetaceans out to ranges of tens of metres from anthropogenic structures over extended time periods. The system provides an efficient means of reducing data volumes to manageable sizes and provides the basis of an effective long-term monitoring tool for identifying and tracking individual animals in discrete locations. From a conservation and management perspective, the approach can be used for monitoring cetacean movements around potentially high-risk anthropogenic activities or structures such as tidal turbines.

## Acknowledgments

The work would also not have been possible without the extensive cooperation of the engineering team at Meygen Atlantis, particularly Lorna Slater, Fraser Johnson and Bruce Mackay. Pre-amplifiers were designed by Mark Johnson. We are also grateful to members of the project steering group for comments on early drafts of this manuscript.

## Software availability

Both the PAMGuard data analysis software and the CRIO data acquisition software are open source and are available at https://sourceforge.net/p/pamguard/svn/HEAD/tree/PamguardJava/ and https://sourceforge.net/p/plabuoy/svn-code/HEAD/tree/ respectively.

## Author Contributions

**Conceptualization:** Douglas Gillespie, Carol Sparling, Gordon Hastie.

**Data curation:** Laura Palmer.

**Formal analysis:** Douglas Gillespie, Laura Palmer, Jamie Macaulay.

**Funding acquisition:** Carol Sparling, Gordon Hastie.

**Investigation:** Douglas Gillespie, Laura Palmer, Jamie Macaulay.

**Methodology:** Douglas Gillespie, Laura Palmer, Jamie Macaulay, Gordon Hastie.

**Project administration:** Carol Sparling, Gordon Hastie.

**Software:** Douglas Gillespie, Jamie Macaulay.

**Supervision:** Douglas Gillespie, Gordon Hastie.

**Writing – original draft:** Douglas Gillespie.

**Writing – review & editing:** Laura Palmer, Jamie Macaulay, Carol Sparling, Gordon Hastie.

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
