## [Decision Letter · Decision Letter 0]

28 Feb 2020

PONE-D-20-02520

Passive acoustic methods for tracking the 3D movements of small cetaceans around marine structures

PLOS ONE

Dear Dr Gillespie,

Thank you for submitting your manuscript to PLOS ONE. After careful consideration, we feel that it has merit but does not fully meet PLOS ONE’s publication criteria as it currently stands. Therefore, we invite you to submit a revised version of the manuscript that addresses the points raised during the review process.

This manuscript is well written, and with a number of minor revisions (listed below), can be acceptable for publication.

We would appreciate receiving your revised manuscript by Apr 13 2020 11:59PM. To enhance the reproducibility of your results, we recommend that if applicable you deposit your laboratory protocols in protocols.io, where a protocol can be assigned its own identifier (DOI) such that it can be cited independently in the future. For instructions see: http://journals.plos.org/plosone/s/submission-guidelines#loc-laboratory-protocols

We look forward to receiving your revised manuscript.

Kind regards,

William David Halliday, Ph.D.

Academic Editor

PLOS ONE

Journal Requirements:

2. In your Methods section, please provide additional location information of the study area, including geographic coordinates for the data set if available.

4. Your ethics statement must appear in the Methods section of your manuscript. If your ethics statement is written in any section besides the Methods, please move it to the Methods section and delete it from any other section. Please also ensure that your ethics statement is included in your manuscript, as the ethics section of your online submission will not be published alongside your manuscript.

Additional Editor Comments (if provided):

Both reviewers and I agree that this paper is well written, and with a small number of relatively minor revisions, can be acceptable for publication. Please address all comments from both reviewers when revising your manuscript. I will note that Reviewer 2 asks why you didn't use whistles for localization. Given that you describe using the PAMGuard whistle detector, this is a relevant query. Why describe using the whistle detector if you aren't using the whistle data here? Additionally, I noted three small errors:

Line 39: the Smith 2000 reference is outdated when discussing increased anthropogenic structures in the marine environment over the past several decades, since that paper is already 20 years old.

Line 242: strange text, likely from your reference management software: "(Error! Reference source not found.)" - this is also noted by Reviewer 1.

Line 267: "an Fourier transform" should be "a fast Fourier transform".

Reviewers' comments:

Reviewer's Responses to Questions

**Comments to the Author**

1. Is the manuscript technically sound, and do the data support the conclusions?

Reviewer #1: Yes

Reviewer #2: Yes

2. Has the statistical analysis been performed appropriately and rigorously? 

Reviewer #1: Yes

Reviewer #2: Yes

3. Have the authors made all data underlying the findings in their manuscript fully available?

Reviewer #1: Yes

Reviewer #2: No

4. Is the manuscript presented in an intelligible fashion and written in standard English?

Reviewer #1: Yes

Reviewer #2: Yes

5. Review Comments to the Author

Reviewer #1: Summary

In this paper the authors describe the implementation and testing of a PAM system aimed at collecting information about 3D movements of harbor porpoise and dolphins around tidal energy generators and, more in general, in the proximity of anthropogenic marine structures which represents a potential risk for the animals during the operation period.

The paper has great merit. The authors made a very good job at introducing the problem of the potential risk of injuries for the cetaceans interacting with anthropogenic underwater structures and at describing the materials and algorithms used to collect data on 3D movements of small cetaceans. In particular, the tuning of the system (Section 3, Offline Analysis) is thoroughly and clearly described and it gives a plausible overview of the strengths and weaknesses of the implemented PAM system.

A noticeable amount of information has been collected on cetaceans (porpoise and dolphin) encounter rates during the first year of operation that should pose the basis for further implementation of the monitoring system.

Maybe, in the conclusions the authors should indicate more explicitly how the information about fine-scale movements of cetaceans in the proximity of the turbines can be practically exploited for preventing cetaceans from possible collisions and consequent injuries.

In conclusion the authors succeeded in demonstrating that data on fine-scale 3D movements of small cetaceans in the range of ~35 meters can be collected and stored for future application in the marine mammals passive acoustic monitoring field for conservation purposes.

Few major and minor issues are reported below.

Major issues

The level of automation and the skill of real-time detection of the system cannot be easily evinced by the text from the beginning.

At line 116 the authors claim that "real-time processing" is conducted on shore using PAMguard software, but at line 124 it is said that the click detector was configured in such a way that the false positive rate is quite high and a post hoc manual screening was necessary to extract cetacean clicks.

From the description reported in Section 3 (“Offline Analysis”) it is clear that a big effort has been made by the authors in tuning the detection/tracking algorithms and in measuring the localization accuracy, but it's not clear if this has been subsequently implemented in the algorithms to make the system autonomous.

Only at the end of the paper ("Conclusions" section, line 395-405) it is asserted that at this moment the system is not totally automated and the developing of an automated encounter detection process is postponed to future studies.

Since the level of automation (i.e. the ability to first recognize true positive source signals and then localize the source) and the response time (real-time aspects) of a PAM system are crucial points for marine mammals conservation purposes, the authors should clarify this point earlier in the text in order to avoid confusion in the reader.

Moreover, the authors should better emphasize how the information about fine-scale movements of cetaceans in the proximity of the turbines can be exploited for preventing cetaceans from possible collisions with moving turbine parts and consequent injuries. For example indicating if this method can be used for implementing a real-time monitoring system equipped with an alarm generation that could break the turbine blades in the presence of one or more individuals.

Minor issues

- the authors should decide the format of the bibliography citations and use it all over the paper: sometimes the square bracket format (for ex [1]) and sometimes the (first author, year) format is used.

- line 61-63

considering that during operation time the turbine produces a high level of acoustic noise centered in a frequency band that partially overlaps with the one of the dolphin whistle (see fig.2), could the system detect dolphin whistle at acceptable percentage of success?

- line 64

been  be

- line 90

... the sound can (be) localised

- line 142-146:

<<therefore, match...="" to="">> The explanation of the division of clicks received at different clusters into 'possible groups' is slightly confusing. It can be evinced that the group assignment criterium is based on time constraint coming from the maximum time delay of a received click at different clusters, nominally max_Tdelay = cluster spacing/ sound speed in water. The authors should better clarify this part to avoid confusion by rephrasing the sentence and/or by supporting the sentence with a simple scheme (like the one in Fig.3)

- line 199-208

<<estimating cartesian="" coordinates..="" errors="" in="">>

Again a simple diagram reporting labeled Cartesian axis and cardinal points could better clarify the sentence.

- line 242

error on reference

Reviewer #2: Review of Gillespie et al. manuscript titled " Passive acoustic methods for 1 tracking the 3D 2 movements of small cetaceans around marine structures "

This paper investigated the development and application of a hydrophone array system for monitoring the three-dimensional movements of cetaceans in the immediate vicinity of a subsea structure. Localization accuracy of this system was also assessed with an artificial sound source at known locations and a refined method of error estimation was also presented. By using this system, high resolution tracking data for animals close to the turbine can be obtained which data can be used to inform stakeholders and regulators on the likely impact of tidal turbines on cetaceans.

My major concern is that it mainly discussed the detection and underwater tracking by using echolocation clicks, and no data are presented for dolphin whistle detections. Additionally, the dolphin detection algorism used for porpoise echolocation detection is not clearly stated.

Some minor revision suggestions was following:

Line 26 localisation should be localization

Line 52 reference (e.g. Johnson and Tyack, 2003) should following the format of the journal

Line 54, same problem as mentioned above

Line 63 reference should change the citing format

Line 65 same problem as mentioned above

Line 112 information of NI-9222, such as manufacture country and company should added

Line 133 Lewis et al., 2018 should modified to the journal required style

Line 138 same problem as mentioned above

 </estimating></therefore,>

6. PLOS authors have the option to publish the peer review history of their article (what does this mean?). If published, this will include your full peer review and any attached files.

Reviewer #1: Yes: Dr. Marco Brunoldi, Ph.D. - Department of Physics - University of Genova - Italy

Reviewer #2: No

---

## [Author Response · Author response to Decision Letter 0]

22 Apr 2020

[This is all in Response to Reviewers.docx]

Comments by editor

To enhance the reproducibility of your results, we recommend that if applicable you deposit your laboratory protocols in protocols.io, where a protocol can be assigned its own identifier (DOI) such that it can be cited independently in the future. 

Response: We don’t see that there is additional detail required for this paper which could go to protocols.io. However, we are considering a submission on a paper we’re writing on porpoise movement using the methods described in this paper. 

Journal Requirements:

Response: We have updated format of the author list / title page and adhered to the file naming requirements. 

2. In your Methods section, please provide additional location information of the study area, including geographic coordinates for the data set if available.

Response: We have added the geographic coordinates (58°39'N 3°08'W) at line 241

Response: Data have been uploaded to https://doi.org/10.17630/de341ca6-6754-43ed-b1ac-a55aae6ccfaa

4. Your ethics statement must appear in the Methods section of your manuscript. If your ethics statement is written in any section besides the Methods, please move it to the Methods section and delete it from any other section. Please also ensure that your ethics statement is included in your manuscript, as the ethics section of your online submission will not be published alongside your manuscript.

Response: The ethics statement has been moved to the start of the Methods section (lines 78-79) and removed from the acknowledgements. 

Additional Editor Comments (if provided):

Both reviewers and I agree that this paper is well written, and with a small number of relatively minor revisions, can be acceptable for publication. Please address all comments from both reviewers when revising your manuscript. I will note that Reviewer 2 asks why you didn't use whistles for localization. Given that you describe using the PAMGuard whistle detector, this is a relevant query. Why describe using the whistle detector if you aren't using the whistle data here? Additionally, I noted three small errors:

Response: I agree that this is a valid query and I have added some text to the discussion to say that very few whistles were detected and that we’ve not attempted to localise them and indeed that accurate localisation may not even be possible for these sounds (lines 396 – 404). 

Line 39: the Smith 2000 reference is outdated when discussing increased anthropogenic structures in the marine environment over the past several decades, since that paper is already 20 years old.

Response: (Line 40) I have replaced this with a reference to Stojanovic TA, Farmer CJQ. The development of world oceans & coasts and concepts of sustainability. Mar Policy. 2013;42: 157–165. doi:10.1016/j.marpol.2013.02.005

Line 242: strange text, likely from your reference management software: "(Error! Reference source not found.)" - this is also noted by Reviewer 1.

Response: I apologise for not adequately checking the generated pdf. It was fine in the submitted word doc. Is now fixed

Line 267: "an Fourier transform" should be "a fast Fourier transform".

Response. Agree. Have changed text (though interestingly, there are now Non FFT algorithms available that are almost as fast as the FFT, so there is really little need these days to restrict ourselves to FFT lengths that are a power of 2. Indeed, the Matlab FFT function will do a FT of any length, so this is something to watch for in future publications). Lines 277 - 278

3. Have the authors made all data underlying the findings in their manuscript fully available?

Reviewer #1: Yes

Reviewer #2: No

 Response: As stated earlier, data have been uploaded to DOI at : 

5. Review Comments to the Author

Reviewer #1: 

Comment:

Maybe, in the conclusions the authors should indicate more explicitly how the information about fine-scale movements of cetaceans in the proximity of the turbines can be practically exploited for preventing cetaceans from possible collisions and consequent injuries.

Response: I have added text to the discussion discussing the possibility of stopping turbines when animals are detected. Our initial priority is to assess whether there is a problem that needs solving rather than to change either turbine or animal behaviour. Implementing a turbine shut-down on detection system would be extremely challenging. (lines 427 to 434).

Major issues

The level of automation and the skill of real-time detection of the system cannot be easily evinced by the text from the beginning. At line 116 the authors claim that "real-time processing" is conducted on shore using PAMguard software, but at line 124 it is said that the click detector was configured in such a way that the false positive rate is quite high and a post hoc manual screening was necessary to extract cetacean clicks.

From the description reported in Section 3 (“Offline Analysis”) it is clear that a big effort has been made by the authors in tuning the detection/tracking algorithms and in measuring the localization accuracy, but it's not clear if this has been subsequently implemented in the algorithms to make the system autonomous.

Only at the end of the paper ("Conclusions" section, line 395-405) it is asserted that at this moment the system is not totally automated and the developing of an automated encounter detection process is postponed to future studies.

Response: These are good points, though in our defence, there are only 8 lines separating the two statements at lines 116 and 124. The real-time processing was used to reduce data volumes by a factor of about 300, then remaining data validated offline. These points are now addressed by a sentence added to the start of the Methods section (lines 75-77): “The system is semi-automatic, with real time detectors reducing amounts of stored data by several orders of magnitude, but operator screening of remaining sounds being required to select and confirm detections from the stored data”

Since the level of automation (i.e. the ability to first recognize true positive source signals and then localize the source) and the response time (real-time aspects) of a PAM system are crucial points for marine mammals conservation purposes, the authors should clarify this point earlier in the text in order to avoid confusion in the reader.

Moreover, the authors should better emphasize how the information about fine-scale movements of cetaceans in the proximity of the turbines can be exploited for preventing cetaceans from possible collisions with moving turbine parts and consequent injuries. For example indicating if this method can be used for implementing a real-time monitoring system equipped with an alarm generation that could break the turbine blades in the presence of one or more individuals.

Response: (We assume the reviewer means “brake” and not “break”). These two points are related and not very relevant to the paper. While I agree that the system as stands is not suitable for real time mitigation, a more fundamental problem than making a detection decision in real time would be the stopping of the turbine. The purpose of our current study is to establish whether or not there is a problem that needs solving, not so solve it. Talking informally with engineers, a braking system that could stop a turbine within a matter of seconds in response to a detection would be a major challenge. I comment on this further in the discussion (lines 417 to 434) and make it clear that the purpose of this system is for study of animals, not for mitigation (indeed, mitigation was deliberately not mentioned once in the original manuscript). 

Minor issues

- the authors should decide the format of the bibliography citations and use it all over the paper: sometimes the square bracket format (for ex [1]) and sometimes the (first author, year) format is used.

Response: Apologies. This has been fixed. 

- line 61-63

considering that during operation time the turbine produces a high level of acoustic noise centered in a frequency band that partially overlaps with the one of the dolphin whistle (see fig.2), could the system detect dolphin whistle at acceptable percentage of success?

Response: We don’t know. This is discussed at line 396 - 404. 

- line 64

been  be

Response: Corrected. (line 62)

- line 90

... the sound can (be) localised

Response: Corrected. 

- line 142-146:

<> The explanation of the division of clicks received at different clusters into 'possible groups' is slightly confusing. It can be evinced that the group assignment criterium is based on time constraint coming from the maximum time delay of a received click at different clusters, nominally max_Tdelay = cluster spacing/ sound speed in water. The authors should better clarify this part to avoid confusion by rephrasing the sentence and/or by supporting the sentence with a simple scheme (like the one in Fig.3)

Response: We don’t think that a diagram will be useful here (there is one in Macaulay 2018) but based on this comment we have amended the text which is now a lot clearer (lines 150 - 160). 

- line 199-208

<>

Again a simple diagram reporting labeled Cartesian axis and cardinal points could better clarify the sentence.

Response: We have added a diagram showing the coordinate system for the errors (new figure 2). Caption at lines 218 to 222 

- line 242

error on reference

Response: Corrected.

My major concern is that it mainly discussed the detection and underwater tracking by using echolocation clicks, and no data are presented for dolphin whistle detections. Additionally, the dolphin detection algorism used for porpoise echolocation detection is not clearly stated.

Response: We have added text at lines 138 - 141 to give further information on the click detection and classification process. 

Some minor revision suggestions was following:

Line 26 localisation should be localization

Response: We have used British English. We find nothing on the journal website requesting US English though are happy to change if requested by the editor. 

Line 52 reference (e.g. Johnson and Tyack, 2003) should following the format of the journal

Line 54, same problem as mentioned above

Line 63 reference should change the citing format

Line 65 same problem as mentioned above

Response to above four points: Apologies. All corrected. 

Line 112 information of NI-9222, such as manufacture country and company should added

Response: Have added National Instruments address. (Line 123)

Line 133 Lewis et al., 2018 should modified to the journal required style

Line 138 same problem as mentioned above

Response to above two points: Apologies. All corrected. 

 All figures have been passed through PACE and uploaded in the correct format.

---

## [Editor Report · Decision Letter 1]

4 May 2020

Passive acoustic methods for tracking the 3D movements of small cetaceans around marine structures

PONE-D-20-02520R1

Dear Dr. Gillespie,

We are pleased to inform you that your manuscript has been judged scientifically suitable for publication and will be formally accepted for publication once it complies with all outstanding technical requirements.

With kind regards,

William David Halliday, Ph.D.

Academic Editor

PLOS ONE
---

## [Editor Report · Acceptance letter]

12 May 2020

PONE-D-20-02520R1 

Passive acoustic methods for tracking the 3D movements of small cetaceans around marine structures 

Dear Dr. Gillespie:

I am pleased to inform you that your manuscript has been deemed suitable for publication in PLOS ONE. Congratulations! Your manuscript is now with our production department. 

With kind regards,

on behalf of

Dr. William David Halliday 

Academic Editor

PLOS ONE